# Factors Associated with School Sports Injury among Elementary and Middle School Students in Shanghai, China

**DOI:** 10.3390/ijerph19116406

**Published:** 2022-05-25

**Authors:** Liyi Ding, Britton W. Brewer, Marcia Mackey, Hao Cai, Jianqiang Zhang, Yudong Song, Qunhui Cai

**Affiliations:** 1Physcial Education College, Shanghai Normal University, 100 Gui Lin Road, Shanghai 200234, China; tych310@shnu.edu.cn; 2Department of Psychology, Springfield College, 263 Alden Street, Springfield, MA 01109, USA; bbrewer@springfieldcollege.edu; 3Department of Physical Education & Sport, Central Michigan University, Mount Pleasant, MI 48859, USA; macke1mj@cmich.edu; 4Physical Education Department, Shanghai Yangguang Foreign Language School, 69 Huancheng Road West, Fengxian District, Shanghai 201499, China; zhangjianqiang1977@hotmail.com (J.Z.); songyudong1970@hotmail.com (Y.S.); caiqunhui@foxmail.com (Q.C.)

**Keywords:** sports injuries, internal and external risk factors, children, adolescents

## Abstract

School sports activity (SSA) is beneficial to gaining and maintaining optimal health among elementary and middle school students but might increase risks for school sports injury (SSI). This cross-sectional study aimed to investigate the incidence and identify risk factors of SSI among Chinese elementary and middle school students in Shanghai. Students in grades 4–5 (elementary) and 7–9 (middle) from three k-12 schools (aged from 9 to 16 years old) in Shanghai selected via the method of cluster random sampling were invited to participate in the study. Information on socio-demography, sleep duration, individual internal and external risk factors, and SSI experiences in the past 12 months was collected. A multivariate logistic regression model was performed to estimate the risk factors of SSI. A total of 1303 participants completed the questionnaires, with an overall SSI incidence rate of 29.5%. Along with boys, elementary school students, and sports team members, students scoring high on internal and external risk factors were at higher risk for SSA. In summary, SSI was prevalent among elementary and middle school students in Shanghai, China, and was associated with several modifiable risk factors. The findings provide insights regarding actions that could be taken to reduce the occurrence of SSI and maximize the benefits of SSA, including improvements in safety education, maintenance of facilities and equipment, and completion of warm-up exercises.

## 1. Introduction

School sports participation is one of the main factors resulting in injuries to children and adolescents at school [1]. According to findings from a 1998 WHO cross-border study on “health behavior of school-age children”, 21.8% of the injuries occurred in children of 11, 13, and 15 age groups, and half of them were injuries in sports and playground activities [2]. A study of students in Montreal found that most injuries were incurred by male students aged 10–14 years, with falls and sports injuries being the most common injury mechanisms [3]. Sheps and Evans reported that most primary school injuries among students in the Vancouver area occurred on the playground, while sports and classroom accidents were the most common causes of injuries among high school students [4]. In recent years, studies have shown that school sports are an important factor in children’s injuries in North America [5,6,7]. This phenomenon can be attributed to the fact that most North American children and adolescents spend almost half of their waking time in school, including five hours of education and two hours in extracurriculars at school. The average number of school days in the United States and Ontario, Canada, were 178 and 185, respectively [8]. Therefore, it is necessary to investigate the related factors resulting in school sports injuries so that effective measures can be taken to solve the problem.

The investigation of school sports injuries can be distinguished from playground injuries. As noted by MacKay [9], playground injuries are most frequently caused by falls from equipment (e.g., monkey bars, slides) or moving objects (e.g., swings). Therefore, only school sports injuries, not playground injuries, are examined within this study. Similarly, gender differences in children’s outdoor play injuries, as reported by Rosen and Peterson [10], are distinct from school sports injuries.

Conclusions from a systematic review [11] support a theoretical framework in which intrinsic and extrinsic factors contribute to the risk for sports injuries in children and adolescents [12,13,14]. Intrinsic risk factors included age, sex, previous injury (nonmodifiable factors) and fitness level, preparticipation sport-specific training, flexibility, strength, joint stability, biomechanics, balance/proprioception, and psychological/social factors (modifiable factors). Extrinsic risk factors included sport played (contact/no contact), level of play (recreational/elite), position played, weather, time of season/time of day (nonmodifiable factors) and rules, playing time, playing surface (type/condition) and equipment (protective/footwear) (modifiable factors). In addition, research [15,16] documented correlations between school sports injuries and such variables as gender, age, school sports team membership, individual safety awareness, the duration of moderate-to-vigorous-intensity physical activities (MVPA), chronic disease, parents’ marital status, wet floor, and rebellious behavior. In these studies [15,16], however, only a limited number of variables were examined, the general relationship of the internal and external risk factors to school sports injuries was not considered, and elementary school students were not assessed. A recent study showed a negative impact of the pandemic on youth physical activity [17]. This negative impact was cross-referenced with school sports injuries.

The purpose of this cross-sectional study was to investigate the incidence rate and identify risk factors for SSI among Chinese elementary and middle school students in Shanghai. Moreover, we want to ascertain the general relationship of internal and external factors to school sports injuries, aiming to provide a basis for the prevention strategies to reduce the occurrence of SSI in elementary and middle school students.

## 2. Materials and Methods

### 2.1. Participants

Using the method of cluster random sampling, we selected the schools by the locations and administrative areas of Shanghai, China. Students in grades 4–5 (i.e., elementary school, aged from 9 to 13 years old) and 7–9 (i.e., middle school, aged from 11 to 16 years old) from three k-12 schools in Shanghai were invited to participate in the study in December 2020. Our exclusion criteria were: (i) students who were unable to participate in normal sports activities and (ii) students who did not agree to engage in the study.

### 2.2. Data Collection

Structured questionnaires were distributed to all eligible students in classes by trained personnel. The questionnaire was self-administered and self-reported, consisting of demographic information, sleep duration, individual internal and external factors of SSI, and SSI experiences in the past 12 months.

The demographic information included name, gender, age, school, study year, height (centimeter), weight (kilogram), sports team member (yes or no), any diagnosed chronic disease/symptom or not (such as heart disease, vision, or hearing disorder), school dormitory residence (yes or no), nearsightedness (yes or no), only child family (yes or no), and their parental marital status.

As for sleep duration (including nap time), the following two questions were asked: (i) “On average, how many hours do you sleep on a typical weekday?”; (ii) “On average, how many hours do you sleep on a typical weekend?”. The average daily time was then generated (using the formula of a sum of duration on a typical weekday multiplied by five and duration on a typical weekend multiplied by two, then divided by seven). Participants were further grouped into five categories (i.e., <6 h, 6 to <7 h, 7 to <8 h, 8 to <9 h, and ≥9 h per day, respectively) based on their average daily sleep duration.

The 19-item Scale for Individual Internal and External Factors of School Sports Injuries (SIIEFSSI) was developed for the current study in accordance with the theory of risk factors for sports injuries [12,13], and the items were adapted from relevant research [15]. (Table 1) Responses to all items are given on a 5-point Likert scale from 1 to 5 (1 = never do, 2 = rarely do, 3 = occasionally do, 4 = often do, 5 = always do). Items #1–8 correspond to external factors and the remaining 11 items (i.e., #9–19) to internal factors. There are 13 items scored positively (i.e., items #1–11, 16, and 18), and 6 items scored negatively (i.e., items #12–15, 17, and 19). 

Adapted from research on sports injuries in children and adolescents [18,19], a school sports injury (SSI) was defined as any physical injuries sustained by a child during P.E. classes, extracurricular sports activities, training sessions, or games, such as contusion, sprain, fracture, etc., resulting in (i) a break in the current activity; and/or (ii) an inability to (fully) participate in the next planned activity; and/or (iii) an inability to attend school the next day; and/or (iv) a need for medical attention (i.e., ranging from first aid to hospitalization) [20]. Furthermore, students reporting injury were requested to provide information about their SSI (including site, causes, injury types, injured body parts, treatment, etc.). The additional injury-related information was collected to validate the SSI outcome measure further.

### 2.3. Procedure

This study was conducted in accordance with the Declaration of Helsinki. Prior to approaching potential participants, approval was obtained from the Shanghai Normal University Ethics Committee (Ethics of SHNU (2020) #35). Explanatory statements and parental consent forms were distributed to 1500 students, with a response rate being 90% (1350), and the questionnaires were subsequently given during school hours to the consenting students in the nominated classes. The purpose of the study was explained orally to the students prior to their completion of the questionnaires, and trained personnel answered questions and provided clarification as needed during the session. Nonetheless, data from 47 (3.48%) students were excluded due to incomplete responses to the questionnaires, resulting in a final valid sample size of 1303.

### 2.4. Statistics

After entering and classifying the data, the scores of negative items were numerically reversed first. Descriptive statistics were calculated to evaluate the characteristics of participants. Continuous and categorical variables were presented as mean (standard deviation, SD) and number (percentage) and tested for between-group differences using independent-sample *t*-tests and chi-square tests, respectively. Then, exploratory factor analysis (EFA) with varimax rotation was performed on SIIEFSSI items for the sample of School B (*n* = 350) using SPSS 22.0 (SPSS Inc., Chicago, IL, USA). Items with factor loadings greater than 0.40 were retained. Confirmatory factor analysis (CFA) in AMOS 23.0 was performed on SIIEFSSI items for the samples of School A + School C (*N* = 968) to determine the goodness-of-fit of the proposed 2-factor model (internal and external factors) [21]. A multivariate logistic regression model was used with the sample of all the three schools (*N* = 1303) to examine all the variables to estimate their risks for SSI, which were expressed in terms of odds ratios (ORs) and 95% confidence intervals (95% CIs). In the multivariate analyses, all the predictor variables were entered into the regression model. A two-sided *p*-value less than 0.05 was considered statistically significant.

## 3. Result

Overall, the present study included 1303 eligible students (52.6% boys and 51.9% elementary school students), with a mean age of 12.10 (SD: 1.85) years. As shown in Table 2, 385 students (29.5%) reported at least one SSI episode in the past 12 months, with significant differences found for age, gender, study year, sports team member, sleep duration, nearsightedness, only child status, internal risk factors, and external risk factors (all *p* < 0.05) between SSI and non-SSI.

EFA eliminated nine items (items #6–11, 16, and 18–19) with factor loadings < 0.04 and yielded a 2-factor (internal and external factors) model with 10 items related to the respective factors. The items that were retained with their means, standard deviations, and factor loadings are presented in Table 3.

The two-factors model for the SIIEFSSI items was further examined with confirmatory factor analysis (CFA) in AMOS 23.0 (Figure 1). As shown in Table 4, the outputs indicated a good fit of the data to the 2-factor [18], with acceptable values for most indices of fit (ꭓ2/df = 6.155, RMSEA = 0.073, CFI = 0.960, GFI = 0.956, AGFI = 0.930, RMR = 0.057), which can be seen in Table 4. The internal consistency for internal risk factors (0.702) and external risk factors (0.918) was acceptable.

Results of the multivariate logistic regression analysis predicting SSI are displayed in Table 5. Middle school students had smaller odds of sustaining SSI than elementary school students, with boys being more likely to experience SSI than girls (OR = 0.539 and 1.359, respectively). The internal and external risk factors were also significant predictors of SSI (OR = 1.255, 95% CI: 1.084–1.454 and OR = 1.142, 95% CI: 1.021–1.277, respectively). In addition, sports team members had a higher risk of SSI than their counterparts not on a sports team (OR = 1.689, 95% CI: 1.231–2.317, *p* < 0.05).

Table 6 summarizes the numbers and consequences of SSI among the 385 students who reported sustaining at least one SSI over the 12 months prior to the administration of the survey stratified by gender. More than one-fifth of the students reporting injury (12.7%, 49/385) indicated that they had experienced at least three SSI events (i.e., multiple injuries). Boys reported experiencing multiple injuries to a greater extent than girls and were more likely than girls to report seeing a doctor or hospitalization after experiencing SSI (13.7% vs. 7.0%, χ^2^ = 4.296, *p* < 0.05). As for other potential consequences of SSI (e.g., immediately stopping the SSA, being absent from the next SSA, and being absent from class the next day), there was no significant difference between the boy’s and girl’s groups (*p* < 0.05).

In addition, we also investigated the main reasons for sports injuries in primary and secondary schools in Shanghai: (i) Inattention (141 students, 36.62%); (ii) Movement technique errors (120 students, 31.17%); (iii) Insufficient preparation activities (100 students, 25.97%); (iv) Problems with site or facilities (42 students, 10.91%); (v) Other aspects (including strain injuries, stampede accidents, carelessness, etc.) (40 students, 10.39%). Since the cause of the injury is a multiple-choice question, the sum of the five causes is greater than 100%.

## 4. Discussion

A total of 1303 students (685 boys and 618 girls) from three k-12 schools in Shanghai, China, participated in our study. The overall self-reported incidence rate of SSI events over a 12-month period was 29.5%. Boys, elementary school students, and sports team members were at elevated risk to report experiencing SSI. In addition, both individual internal and external risk factors were associated with increased risk for reporting an SSI.

The results of our study showed that 385 primary and secondary school students (29.5%) reported experiencing at least one SSI in the past year, which was consistent with the proportion of middle school students with sports injuries (25.1%) in a related study [15]. Previous studies [15,22] showed that factors for sports injuries mainly include gender, age, study year, school sports team membership, personal safety awareness, duration of moderate to vigorous physical activity, chronic disease, parental marital status, wet ground, rebellious behavior, and school dormitory residence. Our study revealed significant differences between injured and uninjured groups in variables such as gender, grade level, and school sports membership, as well as internal and external risk factors (*p* < 0.05). The finding of elevated SSI risk for boys in the current study is consistent with that demonstrated across a variety of children’s physical activity contexts ranging from play to competitive sport [10,15,22,23]. Because the factors that may contribute to SSI differ across student locations, ages, genders, family compositions, etc., preventive interventions should be implemented in a targeted manner according to the characteristics of the students and their situation.

The current study differs from previous studies in several respects. First, we designed scales to assess internal and external factors and documented associations between both internal and external risk factors and SSI. Previous studies were mainly focused on identifying potential individual risk factors. Secondly, we compared the occurrence of sports injuries between primary and secondary school students and found that primary school students were more likely to become injured than middle school students. This difference may be due to a relative lack of safety awareness, sports experience, and/or self-protection knowledge among elementary school students. Previous studies have usually examined middle school students as research participants, so our study will enrich knowledge in this field by including primary school students. Third, we also found that the current primary and secondary school students in Shanghai, China, differed by gender only for the highest degree of injury severity (i.e., seeing a doctor or being hospitalized). Potential explanations for this difference include gender differences in sports choice and sensation-seeking [15,16].

From previous studies, we know that sports injuries in children and adolescents are related to intrinsic and extrinsic risk factors and that both intrinsic and extrinsic risk factors include variable and immutable risk factors [11,24]. Although the main purpose of our research was to investigate the current incidence of school sports injuries and related factors, our goal is to prevent and reduce the occurrence of school sports injuries. Toward those ends, we intend to implement a theory-based preventive intervention program focused on modifying changeable risk factors. Preventive efforts are of particular importance, given advocacy for reversing the adverse impact of the pandemic on youth physical activity [17].

Although we considered several factors theoretically related to school sports injuries during the development of SIIEFSSI, including venue/facilities, safety awareness, practice time, and safety knowledge, the results of the exploratory analyses suggest that safety knowledge and safety awareness were the main internal risk factors and that venue/facilities were the main external risk factors associated with SSI. These findings suggest several directions for future research on school sports injuries. First, safety education should be enhanced to improve students’ safety awareness. Second, schools should conduct regular inspections and maintain venue and facilities and should replace and repair equipment with potential safety hazards in a timely manner. Third, attention should be paid to improving the effectiveness of warm-up exercises to better prevent the occurrence of school sports injuries.

The strengths of our study are (i) developing a scale for assessing internal and external risk factors for SSI (i.e., the SIIEFSSI), (ii) verifying positive associations between both internal and external risk factors and school sports injuries, and (iii) including a sample of elementary school students in the study.

Our study has several limitations that should be considered when interpreting the results. First, the study featured a cross-sectional design and relied upon retrospective and self-report data. Because the SSI data were based on the recollections of primary and secondary school students over a 12-month period, the responses may have been susceptible to forgetting and recall bias. Similarly, the height and weight data were self-reported rather than measured, which might have affected the accuracy of BMI scores. Second, due to a lack of data on school sports exposure time, it was not possible to calculate the injury rate in the way it is customarily reported. Third, due to a limitation of funds and personnel, this study only recruited primary and middle school students from three schools in Shanghai, which may limit the generalizability of the current findings.

There are several logical directions for future research. First, the current findings should be replicated using prospective longitudinal data. Second, to enhance the generalizability of the current results, a larger number of schools should be included in future investigations. Third, only a limited number of internal and external risk factors were included in the development of the SIIEFSSI. Additional potential internal and external risk factors could be examined in future research to enhance the predictive power of the scale.

## 5. Conclusions

SSI was prevalent among elementary and middle school students in Shanghai, China. Elementary school students and students who were sports team members were at greater risk for SSI than middle school students and students who were not sports team members, respectively. In addition, both internal and external risk factors were positively associated with SSI. The results of the current study provide a rationale for implementing preventive interventions to reduce the risk for SSI and maximize the benefits of SSA in elementary and middle school students if they train groupwise.

## Figures and Tables

**Figure 1 ijerph-19-06406-f001:**
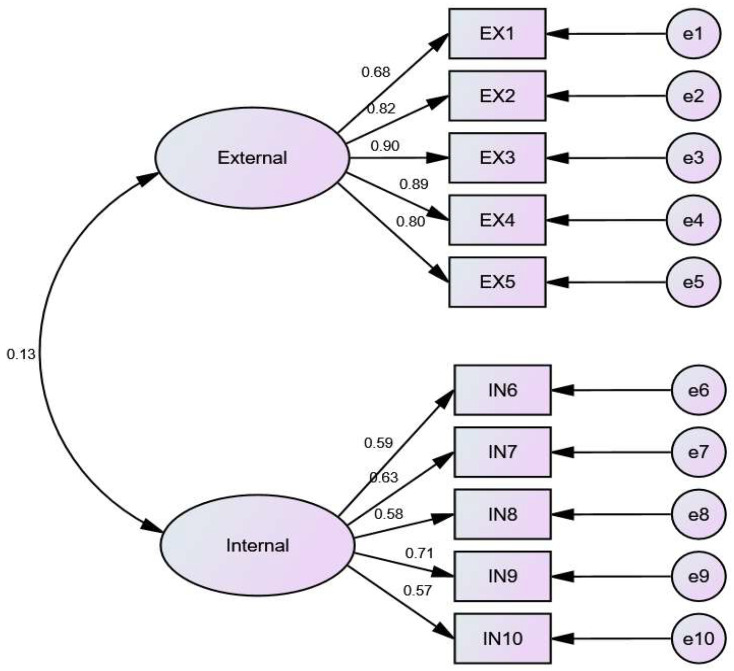
CFA Output for SIIEFSSI, as explained in Table 3.

**Table 1 ijerph-19-06406-t001:** Items for Internal and External Factors resulting in School Sports Injuries (SSI).

Items	Domain	Factors	Direction
1. I will perform physical activities on slippery ground.	External	Facility	Positive
2. I will perform physical activities on uneven ground.	External	Facility	Positive
3. I will perform physical activities in places with poor light.	External	Facility	Positive
4. I will perform physical activities in small venues.	External	Facility	Positive
5. I will perform physical activities in crowded venues.	External	Facility	Positive
6. I will do outdoor physical activities during hot weather.	External	Playing Time	Positive
7. I will do outdoor physical activities when the weather is very cold.	External	Playing Time	Positive
8. I will be physically active when I am sick.	External	Playing Time	Positive
9. I will run and chase on campus.	Internal	Safety Awareness	Positive
10. I will wear necklaces and other accessories during sports activities.	Internal	Safety Awareness	Positive
11. I will wear glasses during sports activities.	Internal	Safety Awareness	Positive
12. When participating in physical activities, I will pay attention to my pulse or heartbeat and other physical conditions.	Internal	Safety Awareness	Negative
13. I will abide by the rules and pay attention to the safety of others when performing physical activities or competitions.	Internal	Safety Awareness	Negative
14. During exercise, if I feel dizzy, I will stop exercising immediately.	Internal	Safety Knowledge	Negative
15. Before exercise or competitions, I will pay attention to maintaining a good sleep to conserve energy.	Internal	Safety Knowledge	Negative
16. Right after the 800-meter race and other strenuous exercises, I will sit down and rest immediately.	Internal	Safety Knowledge	Positive
17. When an ankle is sprained, I will take quick measures, such as applying an ice pack to the wound.	Internal	Safety Knowledge	Negative
18. I will play sports on facilities or equipment with potential safety hazards.	Internal	Safety Awareness	Positive
19. During the exercise, I will drink water regularly.	Internal	Safety Knowledge	Negative

**Table 2 ijerph-19-06406-t002:** Socio-demographic characteristics and other factors as a function of school sports injury (SSI) status.

Characteristics ^1^	All (*N* =1303) *n* (%)	Non-SSI (*N* = 918) *n* (%)	SSI (*N* = 385) *n* (%)	χ^2^/*t* *	*p*-Value ^3^
Gender				8.949	0.003 **
Boy	685 (52.6)	458 (66.9)	227 (33.1)		
Girl	618 (47.4)	460 (74.4)	158 (25.6)		
Study Year				74.993	0.000 ***
Elementary school	676 (51.9)	405 (59.9)	271 (40.1)		
Middle school	627 (48.1)	513 (81.8)	114 (18.2)		
Parental marital status				3.383	0.184
Married	1172 (89.9)	831 (70.9)	341 (29.1)		
Divorced/Separated	122 (9.4)	83 (68.0)	39 (32.0)		
Others	9 (0.7)	4 (44.4)	5 (55.6)		
Sports team member				13.086	0.000 ***
No	1078 (82.7)	782 (72.5)	296 (27.5)		
Yes	225 (17.3)	136 (60.4)	89 (39.6)		
Sleep duration				28.644	0.000 ***
<7 h/d	130 (10.0)	101 (77.7)	29 (22.3)		
7–<8 h/d	273 (21.0)	221 (81.0)	52 (19.0)		
8–<9 h/d	406 (31.2)	281 (69.2)	125 (30.8)		
≥9.00 h/d	494 (37.8)	315 (63.8)	179 (36.2)		
Chronic disease/symptom				3.349	0.067
No	1282 (98.4)	907 (70.7)	375 (29.3)		
Yes	21 (1.6)	11 (52.4)	10 (47.6)		
Living in a school dormitory				1.349	0.246
No	1292 (99.2)	912 (70.6)	380 (29.4)		
Yes	11 (0.8)	6 (54.5)	5 (45.5)		
Nearsightedness				10.758	0.001 **
No	606 (46.5)	400 (66.0)	206 (34.0)		
Yes	697 (53.5)	518 (74.3)	179 (25.7)		
Only child family				7.639	0.006 **
No	760 (58.3)	513 (67.5)	247 (32.5)		
Yes	543 (41.7)	405 (74.6)	138 (25.4)		
Age (x ± s, years)	12.10 ± 1.85	12.38 ± 1.86	11.43 ± 1.65	8.717	0.000 ***
BMI ^2^ (x ± s, kg/m^2^)	19.56 ± 4.50	19.67 ± 4.46	19.34 ± 4.93	1.202	0.230
Internal risk factors	2.02 ± 0.85	1.99 ± 0.85	2.11 ± 0.87	−2.400	0.017 *
External risk factors	1.90 ± 1.10	1.84 ± 1.04	2.05 ± 1.20	−3.126	0.002 **

^1^ Categorical variables (all variables except for age and BMI) were tested by chi-square tests, while continuous variables (age, BMI, internal and external risk factors) were tested by independent-sample *t*-tests; ^2^ BMI, body mass index; ^3^ * *p* < 0.05, ** *p* < 0.01, *** *p* < 0.001.

**Table 3 ijerph-19-06406-t003:** EFA results for SIIEFSSI internal and external risk factor (*N* = 335).

Items	Mean	SD	FL
EX1. I will perform physical activities on slippery ground.	3.94	1.416	0.797
EX2. I will perform physical activities on uneven ground.	4.09	1.316	0.858
EX3. I will perform physical activities in places with poor light.	4.09	1.241	0.871
EX4. I will perform physical activities in small venues.	4.16	1.260	0.856
EX5. I will perform physical activities in crowded venues.	4.20	1.199	0.836
IN6. When participating in physical activities, I will pay attention to my pulse or heartbeat and other physical conditions.	3.38	1.332	0.627
IN7. I will abide by the rules and pay attention to the safety of others when performing physical activities or competitions.	4.27	1.058	0.662
IN8. During exercise, if I feel dizzy, I will stop exercising immediately.	3.83	1.291	0.625
IN9. Before exercise or competitions, I will pay attention to maintaining a good sleep to conserve energy.	4.12	1.101	0.792
IN10. When an ankle is sprained, I will take quick measures, such as applying an ice pack to the wound.	3.77	1.326	0.640

External risk factors: EX1, EX2, EX3, EX4, and EX5 (5 items; Cronbach α = 0.918); Internal risk factors: IN6, IN7, IN8, IN9, and IN10 (5 items; Cronbach α = 0.702).

**Table 4 ijerph-19-06406-t004:** CFA model: Goodness-of-fit statistics.

Model Fit Index	Value Considered Good Fit	Model Fit Value	Indication of Fit
χ^2^/df	Value lower than 3	6.155	Acceptable
RMSEA	0.03–0.08	0.073	Reasonable
CFI	Value > 0.9	0.960	Perfect
AGFI	Value > 0.9	0.930	Good
GFI	Value > 0.9	0.956	Perfect
NFI	Value > 0.9	0.953	Perfect
RMR	Value < 0.05	0.057	Acceptable
IFI	Value > 0.9	0.960	Perfect

**Table 5 ijerph-19-06406-t005:** Multivariate logistic regression to estimate risks for school sports injury among elementary and middle school students in Shanghai, China (*N* = 1303).

Characteristics	β	Wald	*p*-Value	Adjusted OR	95% CI
Gender					
Girl				1	
Boy	0.312	5.530	0.019 *	1.367	1.053–1.773
Study year					
Elementary				1	
Middle	−0.670	5.455	0.020 *	0.512	0.292–0.898
Parental marital status					
Married				1	
Divorced/Separated	0.366	2.703	0.100	1.443	0.932–2.233
Others	1.356	3.663	0.056	3.882	0.968–15.568
Sport team					
No				1	
Yes	0.524	10.569	0.001 **	1.689	1.231–2.317
Sleep Duration					
<7 h/d				1	
7–<8 h/d	−0.227	0.688	0.407	0.797	0.466–1.363
8–<9 h/d	0.122	0.226	0.634	1.130	0.683–1.870
≥9.00 h/d	0.100	0.143	0.705	1.105	0.658–1.855
Chronic disease/symptom					
No				1	
Yes	0.716	2.393	0.122	2.047	0.826–5.073
School dormitory residence					
No				1	
Yes	0.243	0.146	0.703	1.275	0.366–4.438
Nearsightedness					
No				1	
Yes	−0.037	0.074	0.785	0.964	0.740–1.255
Only child					
No					
Yes	−0.047	0.114	0.736	0.954	0.727–1.252
Age (years)	−0.120	2.337	0.126	0.887	0.760–1.034
BMI ^1^ (kg/m^2^)	0.004	0.095	0.758	1.004	0.977–1.032
External Factors	0.133	5.419	0.020 *	1.142	1.021–1.277
Internal Factors	0.227	9.187	0.002 **	1.255	1.084–1.454

* *p* < 0.05; ** *p* < 0.01. ^1^: BMI, body masss index.

**Table 6 ijerph-19-06406-t006:** Numbers and consequences of school sports injury (SSI) between boys and girls. (*N* = 385).

Characteristics	All (*N* = 385) *N*(%)	Gender		χ^2^	*p*-Value
		Boys (*N* = 223) *n* (%)	Girls (*N* = 156) *n* (%)		
Number of SSI				0.114	0.945
1	250 (64.9)	146 (64.3)	104 (65.8)		
2	86 (22.3)	52 (22.9)	34 (21.5)		
≥3	49 (12.7)	29 (12.8)	20 (12.7)		
Consequences of SSI					
Immediately stop the SSA				2.626	0.105
No	170 (44.2)	108 (47.6)	62 (39.2)		
Yes	215 (55.8)	119 (52.4)	96 (60.8)		
Absent from the next SSA				0.453	0.501
No	299 (77.7)	179 (78.9)	120 (75.9)		
Yes	86 (22.3)	48 (21.1)	38 (24.1)		
Class absence next day				1.982	0.159
No	343 (89.1)	198 (87.2)	145 (91.8)		
Yes	42 (10.9)	29 (12.8)	13 (8.2)		
See a doctor or hospitalization				4.296	0.038 *
No	343 (89.1)	193 (86.3)	147 (93.0)		
Yes	42 (10.9)	31 (13.7)	11 (7.0)		

* *p* < 0.05.

## Data Availability

The data presented in this study are available on request from the corresponding author.

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
