# Peer review of "Factors Associated with School Sports Injury among Elementary and Middle School Students in Shanghai, China"

_ijerph, 2022, doi:10.3390/ijerph19116406_

Round 1

Reviewer 1 Report

The authors are fully aware of the limitations of the study. Additional potential internal and external risk factors could have been examined in the present study to enhance the predictive power of the scale.

Author Response

The authors are fully aware of the limitations of the study. Additional potential internal and external risk factors could have been examined in the present study to enhance the predictive power of the scale.

  • Thank you for the time and effort you spent reviewing and commenting on our manuscript.

Reviewer 2 Report

Dear Authors, your paper is very professionally written and truly clear. My compliments.

You find a few detailed remarks below.

Details, typos, suggestions for improvements, and discussions

For your reading convenience, I sometimes propose words in bold.

  1. Line 65, a reference to “these studies” is missing. Is it [10 – 14]? Please insert.

  2. Lines 68 – 72, have one too-long sentence. I propose to split it after the name ‘Shanghai’, by changing “and make certain the general …” into “Also, we want to ascertain the general …”.

  3. Between Lines 99 – 100, in the last column, you use “Reverse”. This, however, is the numerical change you must make. If the direction is ‘negative’ then you must reverse the numbers.
    Concluding: I propose to replace in the Direction column the term ‘reverse’ with ‘negative’.

  4. Line 102 is one of the lines without a blank between the character ‘h’ and the bracket ‘[‘. This omission happens often in the text, see for instance line 237. Please amend.

  5. Lines 104 – 105, to me it appears that here the terms ‘internal’ (line104) and ‘external’ (line 105) are abusively interchanged.

  6. Line 129, I propose to change “. . . negative items were reversed first.” Into “. . . negative items were numerically reversed first.”

  7. Line 158, can "and their means" be expressed better by "with their means"?

  8. Line 166, the reference ‘[18]’ comes after a listing of column 3 of Table 4. I propose – for clarity – to move ‘[18]’ to line 165, or 164 in the text.

  9. Line 170, the capture could be extended by mentioning “Output for SIIEFSSI, as explained in Table 3.

  10. Between Lines 172 – 173, the capture of table 4 ends with a star: * without explanation.

  11. Line 215, it seems appropriate to move the references [13,19] to line 212: “Previous studies [13,19] have shown …”

  12. Lines 278 - 279, the sentence does read clearly. The word “properly” should be deleted, I think.

  13. Lines 281, the sentence should be extended, I think: “. . . and middle school students, if they train groupwise.”

Author Response

  1. Line 65, a reference to “these studies” is missing. Is it [10 – 14]? Please insert.
  • We have cited sources 15 and 16 in support of “these studies.”

  1. Lines 68 – 72, have one too-long sentence. I propose to split it after the name ‘Shanghai’, by changing “and make certain the general …” into “Also, we want to ascertain the general …”.
  • The sentence was revised as suggested.

  1. Between Lines 99 – 100, in the last column, you use “Reverse”. This, however, is the numerical change you must make. If the direction is ‘negative’ then you must reverse the numbers. Concluding: I propose to replace in the Direction column the term ‘reverse’ with ‘negative’.
  • We have replaced the term ”reverse” with “negative.”

  1. Line 102 is one of the lines without a blank between the character ‘h’ and the bracket ‘[‘. This omission happens often in the text, see for instance line 237. Please amend.
  • We have inserted a blank on line 102 & line 237, respectively. The spacing prior to all brackets was checked for consistency and adjusted accordingly.

  1. Lines 104 – 105, to me it appears that here the terms ‘internal’ (line104) and ‘external’ (line 105) are abusively interchanged.
  • Thank you for catching this error. We have switched the two terms as appropriate.

  1. Line 129, I propose to change “. . . negative items were reversed first.” Into “. .. negative items were numerically reversed first.”
  • We have made the suggested change.

  1. Line 158, can "and their means" be expressed better by "with their means"?
  • We replaced “and” with ”with” as suggested.

  1. Line 166, the reference ‘[18]’ comes after a listing of column 3 of Table 4. I propose – for clarity – to move ‘[18]’ to line 165, or 164 in the text.
  • We have moved the reference in question (now [21]) to line 165 behind the 2-factor as suggested.

  1. Line 170, the capture could be extended by mentioning “Output for SIIEFSSI, as explained in Table 3.
  • As suggested, we have added “as explained in Table 3” after “Output for SIIEFSSI.”

  1. Between Lines 172 – 173, the capture of table 4 ends with a star: * without explanation.
  • We have deleted the asterisk (i.e., ”*”)

  1. Line 215, it seems appropriate to move the references [13,19] to line 212: “Previous studies [13,19] have shown …”
  • Moved [13, 19] to line 212 behind Previous studies, as suggested.

  1. Lines 278 - 279, the sentence does read clearly. The word “properly” should be deleted, I think.
  • Sorry, we cannot find the word “properly” in our manuscript.

  1. Lines 281, the sentence should be extended, I think: “. . . and middle school students, if they train groupwise.”
  • We have added the sentence “if they train groupwise.”

Reviewer 3 Report

This is an interesting and important study. I am glad a got a chance of reviewing it and hopefully my comments will be helpful for the authors. 

Title is informative and gives an idea of what the article is about. 

Abstract is generally well-written but could be enhanced, specifically in the last lines where you could be more specific about findings and possibly about some recommendations. 

Introduction definitely needs strengthening as the topic that you touched in your study is multi-factoral and complex, but at the same time it is a big medical, which may not be so interesting for those who should be interested the most - for the PE teachers. It would be good if you could look at the issue from a broader perspective. Therefore, you could look into some factors which clearly influence the occurrence like gender (look for example into "Gender differences in children's outdoor play injuries: A review and an integration'). Also, I advice you have a look at a relation of free play and % of incidents of injuries (look for example into 'Playground injuries to children'). 

Another issue that I feel that the Introduction has been lacking is the problem of declining level of physical fitness of youth (especially in 'Comparison of Physical Activity Levels in Youths before and during a Pandemic Lockdown') and growing prevalence of obesity which impact the growth of movement control problems (look into "Injuries, Pain, and Catastropizing level in Gymnasts: a retrospective analysis of a cohort of Spanish Athletes).

Methodology of the research is well-described which allows for repetition of the research procedure by potential others. Research sample was selected in a reasonable way and characterised clearly. So were the research tools, and good that authors provided all the phases of establishing the reliability of the tools. 

Results are presented in a neat manner, and the most important data has been clearly described in the body of this section. Tables are also neat and one can easily read them through. 

Discussion is also well-written and authors discuss their findings against the ones of other authors, but I would suggest referring here to the issues and articles suggested in Introduction section. This will certainly broaden the context and add value to Discussion (especially that the number of references is not impressive). 

I also appreciate section on limitations - I can see that the authors have a good sense of has been done and what to do next with their suggestions for further research.  

Good luck with your revision.  

Author Response

  1. Abstract is generally well-written but could be enhanced, specifically in the last lines where you could be more specific about findings and possibly about some recommendations.
  • As suggested, we have expanded the final lines of the Abstract.

  1. Introduction definitely needs strengthening as the topic that you touched in your study is multi-factoral and complex, but at the same time it is a big medical, which may not be so interesting for those who should be interested the most - for the PE teachers. It would be good if you could look at the issue from a broader perspective. Therefore, you could look into some factors which clearly influence the occurrence like gender (look for example into "Gender differences in children's outdoor play injuries: A review and an integration'). Also, I advise you have a look at a relation of free play and % of incidents of injuries (look for example into 'Playground injuries to children'). Another issue that I feel that the Introduction has been lacking is the problem of declining level of physical fitness of youth (especially in 'Comparison of Physical Activity Levels in Youths before and during a Pandemic Lockdown') and growing prevalence of obesity which impact the growth of movement control problems (look into "Injuries, Pain, and Catastropizing level in Gymnasts: a retrospective analysis of a cohort of Spanish Athletes).
  • As suggested, we have modified the Introduction to include information on gender differences in play injuries, playground injuries, and pandemic effects on youth physical activity.

  1. Methodology of the research is well-described which allows for repetition of the research procedure by potential others. Research sample was selected in a reasonable way and characterized clearly. So were the research tools, and good that authors provided all the phases of establishing the reliability of the tools.
  • Thank you for your positive feedback.

  1. Results are presented in a neat manner, and the most important data has been clearly described in the body of this section. Tables are also neat and one can easily read them through.
  • Thank you for your positive feedback.

  1. Discussion is also well-written and authors discuss their findings against the ones of other authors, but I would suggest referring here to the issues and articles suggested in Introduction section. This will certainly broaden the context and add value to Discussion (especially that the number of references is not impressive).
  • We have added 5 references to the manuscript and returned in the Discussion to the issues of gender and pandemic effects on youth physical activity raised in the Introduction.
